# Short-Term Effects of PM_10_, NO_2_, SO_2_ and O_3_ on Cardio-Respiratory Mortality in Cape Town, South Africa, 2006–2015

**DOI:** 10.3390/ijerph19138078

**Published:** 2022-06-30

**Authors:** Temitope Christina Adebayo-Ojo, Janine Wichmann, Oluwaseyi Olalekan Arowosegbe, Nicole Probst-Hensch, Christian Schindler, Nino Künzli

**Affiliations:** 1Department of Epidemiology and Public Health, Swiss Tropical and Public Health Institute, Kreuzstrasse 2, Allschwil, 4123 Basel, Switzerland; oluwaseyiolalekan.arowosegbe@swisstph.ch (O.O.A.); nicole.probst@swisstph.ch (N.P.-H.); christian.schindler@swisstph.ch (C.S.); nino.kuenzli@swisstph.ch (N.K.); 2Faculty of Medicine, University of Basel, 4056 Basel, Switzerland; 3Faculty of Health Sciences, School of Health Systems and Public Health, University of Pretoria, Pretoria 0002, South Africa; janine.wichmann@up.ac.za

**Keywords:** multi-pollutant, air pollution, mortality, harvesting, South Africa

## Abstract

Background: The health effect of air pollution is rarely quantified in Africa, and this is evident in global systematic reviews and multi-city studies which only includes South Africa. Methods: A time-series analysis was conducted on daily mortality (cardiovascular (CVD) and respiratory diseases (RD)) and air pollution from 2006–2015 for the city of Cape Town. We fitted single- and multi-pollutant models to test the independent effects of particulate matter (PM_10_), nitrogen dioxide (NO_2_), sulphur dioxide (SO_2_) and ozone (O_3_) from co-pollutants. Results: daily average concentrations per interquartile range (IQR) increase of 16.4 µg/m^3^ PM_10_, 10.7 µg/m^3^ NO_2_, 6 µg/m^3^ SO_2_ and 15.6 µg/m^3^ O_3_ lag 0–1 were positively associated with CVD, with an increased risk of 2.4% (95% CI: 0.9–3.9%), 2.2 (95% CI: 0.4–4.1%), 1.4% (95% CI: 0–2.8%) and 2.5% (95% CI: 0.2–4.8%), respectively. For RD, only NO_2_ showed a significant positive association with a 4.5% (95% CI: 1.4–7.6%) increase per IQR. In multi-pollutant models, associations of NO_2_ with RD remained unchanged when adjusted for PM_10_ and SO_2_ but was weakened for O_3_. In CVD, O_3_ estimates were insensitive to other pollutants showing an increased risk. Interestingly, CVD and RD lag structures of PM_10_, showed significant acute effect with evidence of mortality displacement. Conclusion: The findings suggest that air pollution is associated with mortality, and exposure to PM_10_ advances the death of frail population.

## 1. Introduction

Since the first half of the twentieth century, the association between ambient air pollution and adverse health outcomes has been documented extensively with historical events such as the smog incidents in Meuse Valley, Belgium (1930), Donora, Pennsylvania (1948), and London (1952). Exposure to high levels of air pollution resulted in increased hospitalization and deaths for cardiovascular and respiratory diseases, particularly in the elderly and those with comorbidities [1]. In a recent assessment, ambient air pollution ranked seventh among modifiable disease risk factors, above other factors such as high cholesterol, household air pollution and alcohol use [2]; in contrast to the other risk factors, air pollution is not easily modifiable at individual levels.

Air pollution consists of a complex heterogeneous mixture of gases and particles, some of which are routinely monitored and referred to as criteria pollutants; these include gaseous pollutants such as nitrogen dioxide (NO_2_), sulphur dioxide (SO_2_), ozone (O_3_) and particles namely-PM_10_ (diameter < 10 µm) and PM_2.5_ (diameter < 2.5 µm). There is no evidence of no-effect-thresholds for these pollutants. In fact, health outcomes have been observed even at levels lower than stringent air quality guidelines promoted by the World Health Organization (WHO) Air Quality Guidelines (AQG) in 2005 [3]. Accordingly, WHO updated its air quality guidelines (AQG) in 2021 based on accumulated evidence from six systematic reviews that considered more than 500 articles on air pollution and health [4]. The new guideline values are lower than in the 2005 edition, for instance, 24-h and annual average values for PM_10_ were reduced to 45 µg/m^3^ from 50 µg/m^3^, and to 15 µg/m^3^ from 20 µg/m^3^, respectively. The guideline value for the NO_2_ annual average had the most substantial reduction from 40 to 10 µg/m^3^. In addition, new guideline values for NO_2_ and ozone were introduced: 25 µg/m^3^ for NO_2_ 24-h mean and 60 µg/m^3^, for O_3_ peak season value—which is the 8-h mean from the highest six-month running-average concentration.

Some of the systematic reviews that contributed to the development of the new AQG reported that a 10 µg/m^3^ rise in the daily mean concentration of PM_10_ was associated with 0.6% (95% CI 0.44–0.77%) and 0.9% (95% CI 0.63–1.2%) increases in daily cardiovascular and respiratory disease mortality, respectively. In addition, 10 µg/m^3^ increases in NO_2_ and O_3_ were associated with an elevated risk of 0.72% (95% CI 0.59–0.85%) and 0.43% (95% CI 0.34–0.52%) in daily all-cause mortality, respectively [5]. Another meta-analysis found a positive association between SO_2_ and all-cause mortality with an estimated additional risk of 0.59% (95% CI: 0.46–0.71%), for a 10 µg/m^3^ increment in 24-h concentration [6].

A total of 263 studies were included in the two reviews, but only one study was from Africa [7]. People in African countries disproportionately experience the burden of outdoor air pollution, however respective health effects are rarely quantified for those countries, with South Africa accounting for the majority of the studies from the region. Thus, the current evidence is largely dominated by single and multi-city studies from North America and Europe [8,9,10,11,12].

Two Multi-Country Multi-City (MCC) Collaborative Research Network studies on air pollution (PM_10_ and O_3_) and mortality have been conducted which included South Africa cities; in both studies it was the only African country [13,14]. Air quality monitoring in other regions of the African continent is not well established, resulting in a lack of data for such studies. In the City of Cape Town, epidemiological time-series studies have shown short-term associations between air pollution and cardio-respiratory disease mortality [7,15], however none of them assessed the independent effects of multiple pollutants. As people are exposed to multiple pollutants in the atmosphere—it is crucial to estimate the independent short-term effects of individual pollutants in multi-pollutant models by including the different pollutants simultaneously. The multipollutant approach is also relevant for the distinction of different sources of air pollution. For instance, PM_10_ is a marker of traffic emissions in addition to other combustion and non-combustion sources. In many cities, NO_2_ is a marker of traffic pollution, while SO_2_ may point to power plant emissions and other fossil fuel combustion sources. In addition, SO_2_ is converted to SO_4_^2−^ in the presence of nitrogen oxides when sunlight is at its brightest and contributes to PM [16]; O_3_, as a secondary pollutant, is a marker of anthropogenic and natural sources [17]. Therefore, using single pollutants as a proxy for complex mixtures of pollutants does not properly account for the health effects caused by the simultaneous exposure to multiple pollutants whose concentrations do not vary proportionally over time [18].

The present study aims to bridge this gap by estimating the associations of daily concentrations of PM_10_, NO_2_, SO_2_, O_3_ with daily mortality due to respiratory and cardiovascular diseases, using both single and multi-pollutant models. In addition, we investigate whether associations differ by age groups, sex, and seasons, and provide detailed effect estimates across a lag period of three weeks.

## 2. Method

### 2.1. Study Area

The City of Cape Town is South Africa’s legislative capital and the capital city for the Western Cape Province. It has a population of 4.6 million people and approximately 1.9 million households in 2020 [19]. A majority of the population are the economically active group within the ages of 15–64 years, which comprise 69.5% of the population.

### 2.2. Outcome and Exposure Data

The study is based on three datasets: cardio-respiratory mortality, air pollution and meteorological data from the City of Cape Town, covering the period 1 January 2006 to 31 December 2015. Mortality data was obtained from Statistics South Africa (StatsSA) after signing a Data User Agreement. The data included daily counts of cardiovascular and respiratory deaths aggregated by age groups (excluding <15 which was not provided by StatsSA) and sex for the study period using the International Classification of Disease, 10th version (ICD–10) (J00-J99) and (I00-I99) to select respiratory and cardiovascular deaths, respectively.

Daily hourly measurements of air pollution data for PM_10_, NO_2_, SO_2_, and O_3_ were obtained for the study period from the City of Cape Town. These are routinely monitored pollutants within the air quality monitoring network of 14 stations. We obtained the daily mean for each station by using a minimum of 18 h measurements (75%) and thereafter calculated the city-level daily mean by taking the averages across stations. Missing values were imputed for stations without measurements provided other stations on the same day were non-missing; otherwise days without measurements at all stations were left as missing. The details and the imputation algorithms are illustrated in an earlier study [20]. 

PM_2.5_ data was not included because this standard only came into effect in 2012, and measurements have been very incomplete until recently. According to South Africa, air quality information system (SAAQIS), monthly reports including statistical description of PM_2.5_ measurements only became available in December 2018, thereafter two stations were added in 2019 (Khayelitsha) and 2020 (Foreshore) [21].

Daily means of hourly temperature and relative humidity data were obtained from South Weather Service (SAWS) for the study period.

### 2.3. Statistical Analysis

We described the distributions of daily death counts, air pollutant levels and meteorological variables using means, standard deviations and percentiles (Table 1). Statistics are presented for the total population and stratified by age group (15–64 and >65), sex (male and female) and season (warm/dry (Nov–Mar) and cold/wet (Apr–Oct)). In addition, we examined the temporal correlation between the air pollutant and meteorological daily means using Spearman correlation coefficients (Table 2).

We investigated the association between daily counts of cardiovascular and respiratory disease (CVD and RD) deaths and air pollutant levels of PM_10_, NO_2_, SO_2_ and O_3_ using quasi-Poisson regression models. The core model was developed without the pollutant variables. Natural splines were used to model time trends and seasonal variations and penalized spline was used for the influences of temperature and relative humidity. An appropriate placement and selection of knots of the spline of calendar time is crucial to avoid overfitting, which may reduce the effects of air pollution, and underfitting, which could bias the results due to confounding by uncontrolled short-term influences. We started with natural splines by placing knots every 45 days over the study period of 3652 days (eight knots per year), which resulted in a sequence of 81 knots. We then iteratively removed the least significant knots until the sum of partial autocorrelations across lags 1 to 28 changed from negative to positive. This process produced 46 knots for CVD and 63 knots for RD. The respective knot sequences were then used in the pollutant models. In addition, we controlled for temperature and relative humidity using penalized splines of their moving average levels over four days (including the day of the event). Finally, “day of the week” and “public holiday” were included as categorical variables with 6 and 1 degrees of freedom, respectively. The basic pollutant model may be written as follows:(1)$$\log\left(E\left[Y_{i}\right]\right)=\alpha +\beta_{1}\times poll+ns\left(time,\;knots=knotseq\right)+s\left(avgtemp03\right)+s\left(avgrh03\right)+\sum \beta_{2i}DOW_{i}+\beta_{3}\times Pubday$$
where $E\left[Y_{i}\right]$ is the expected number of deaths on day *given the predictor variables,*
poll is the two-day moving average of the respective pollutant, time is time in days, starting at 1 January 2006, avgtemp03 and avgrh03 are the averages of temperature and relative humidity across lags 0 to 3, $DOW_{i}$ (*i* = 1, …, 6) are indicator variables for the six days of the week other than Sunday, and Pubday is an indicator variable for government recognized public holidays. The Greek letters stand for regression parameters, ns for “natural spline” and s for “penalized spline function”. The moving averages of two-day and three-day for pollutant and meteorological covariates are typically used in studies investigating short-term effects.

Two-, three- and four-pollutant models were run for all respective combinations among the four pollutants. In addition to the overall analysis, the models included interaction terms to assess effect modification by sex (male vs. female), age groups (15–64 and >65) and seasons (warm/dry vs. cold/wet months).

To estimate the cumulative effects over 21 lags, we used a cross basis function for each pollutant within a distributed lag non-linear (DLNM) framework [22]. A linear exposure-response function with a natural cubic spline for the lag weights were specified in addition to placing knots at lags 2, 5, and 9. We again considered lags 0 to 3 for temperature and relative humidity and defined the crossbasis for the two variables involving natural splines for *argvar* and *arglag* with 5 and 3 degrees of freedom, respectively.

All statistical analyses were conducted using R statistical software version 4.0.3 using mgcv, splines and dlnm packages.

## 3. Results

There were a total of 54 356 CVD deaths and 20 376 RD deaths between 1 January 2006 and 31 December 2015, as reported in Table 1. The mean number of deaths per day was 15 for CVD and 6 for RD; mean daily levels of air pollutants were 30.4 µg/m^3^, 16.6 µg/m^3^, 10.5 µg/m^3^, and 33.1 µg/m^3^ for PM_10_, NO_2_, SO_2_ and O_3_, respectively. The 2021 WHO air quality guideline 24-h values were exceeded on 497 (13.6%) of the 3652-day study period for PM_10_ (>45 µg/m^3^), 501 (13.7%) days for NO_2_ (>25 µg/m^3^), and 196 (5.4%) days for SO_2_ (>40 µg/m^3^). There was high variability in the pollutant levels during the study period. For instance, the minimum level for PM_10_ was 6.6 µg/m^3^, while the maximum level was 98.6 µg/m^3^. The daily average temperature and relative humidity were 17.4 °C and 69.8%, respectively. Table 2 shows weak but positive pairwise correlations for the daily mean concentrations of air pollutants, with the exception of SO_2_ and O_3_ (r = −0.11), whilst temperature and relative humidity were negatively correlated (r = −0.41). When stratified by season, the strength of the correlation between PM_10_ and NO_2_ increased moderately (r = 0.54) during the cold/dry season compared to the weaker correlation (r = 0.29) observed in the warm/dry season. In addition, ozone was negatively correlated with other pollutants during the cold/dry season. Correlation results by season for the exposure variables are shown in the Appendix A.

The estimated relative risks and 95% confidence intervals for deaths due to CVD and RD per interquartile range (IQR) increase of the 2-day moving average (lag 0–1) concentration of the four pollutants are presented in Table 3. The RR estimates for the corresponding 10 µg/m^3^ increments are provided in the Appendix A.

We observed positive and statistically significant associations between PM_10_ and CVD deaths. The overall risk of CVD death increased by 2.4% (95% 0.9–3.9%) per IQR increase of 16.4 µg/m^3^ in PM_10_ exposure. In subgroup analyses, higher associations were observed for persons aged ≥65 years and females with risk increments of 3.3% (95% CI 1.4–5.2%) and 3.2% (1.2–5.3%), respectively. However, these results did not significantly differ from the ones in the complementary subgroup with *p*-values 0.26 and 0.72 for age and gender, respectively. NO_2_ was also positively associated with CVD deaths, with an overall increased risk of 2.2% per IQR increase of 10.7 µg/m^3^. This association was similar for females and elderly (age ≥ 65 years) except for age 15–64 and males where the relative increases in risk were 0.1% (95% CI −1.9% to 4.1%) and 1.9% (95% CI −0.7 to 4.5%), respectively. An IQR increase of 6 µg/m^3^ in the two-day mean of SO_2_ was associated with an increased risk of CVD-death by 1.4% (95% CI 0–2.8%); results in subgroups were not statistically significant with the exception of the risk increase by 2.2% (95% CI 0.3–4.1%) in males. Finally, for ozone the estimated overall increase in the risk of CV-death was 2.5% (95% CI 0.2–4.8%) per IQR increase of 15.6 µg/m^3^. There were no statistically significant differences in the estimates across strata of age or gender. For example, the largest differences were seen for ozone, where the risk increment was higher in age ≥65 (2.9%; 95% CI 0–5.9%) versus age 15–64 (1.9%; 95% CI −1.8–5.7%), and males (3.5%; 95%CI 0.2–6.9%) versus females (1.8%; 95% CI −1.3–5%).

For RD mortality, we observed the strongest associations with NO_2_. Overall, there was an estimated risk increase of 4.5% (95% CI 1.4–7.6%) for an IQR increase of 10.7 µg/m^3^ in the 2-day mean of NO_2_. In the age group ≥ 65 years, the respective estimate was 4.9% (95% CI 0.7–9.3%) and in females it was 4.9% (95% CI 0.4–9.6%). The association of daily RD-deaths with the two-day means of SO_2_ was also positive overall, but did not reach statistical significance, while there was almost no association with PM_10_ and O_3_.

The two-, three- and four-pollutant models presented in Figure 1 show an independent positive association between PM_10_ and CVD mortality. The percentage change associated with an IQR increment in PM_10_ increased from 2.4% to 2.8% after adjusting for O_3_ and to 3% after adjusting for SO_2_ and O_3_, although it decreased to 1.9% when we controlled for NO_2_. Adjustment for all three co-pollutants reduced the magnitude of the association to 2% and the statistical significance was no longer attained. The association between CVD mortality and NO_2_ was substantially reduced after adjusting for PM_10_, with a decrease in the risk increment from 2.2% to 1%, but the association remained rather stable in the multipollutant models not including PM_10_. Estimates for SO_2_ became weaker and non-significant in all multi-pollutant models and even turned negative after simultaneous adjustment for PM_10_ and O_3_. Conversely, the association between O_3_ and CVD mortality remained rather unaffected by the inclusion of other pollutants. In summary, associations of CVD-mortality with PM_10_ and NO_2_ were weakened when both pollutants were included together, while associations with O_3_ were not sensitive to the inclusion of other pollutants.

Consistent with the single pollutant model, only NO_2_ had a positive and significant association with RD mortality after adjusting for other pollutants, as seen in Figure 2. The estimated overall risk increment increased from 4.5% to 5.8% after adjusting for PM_10_, while it was slightly reduced to 4.3% after adjusting for SO_2_ and the estimates decreased and became statistically insignificant when adjusted for O_3_. In the three-pollutant model with PM_10_ and SO_2_, the estimate remained high at 5.5%. Adjusting for all three co-pollutants resulted in a slightly attenuated estimate of 4% which was no longer statistically significant. Conversely, the risk estimates for PM_10_ and O_3_ in the multi-pollutant models confirm the corresponding null findings of the single pollutant models, with some random fluctuation of estimates across the various multi-pollutant models. Similar to the overall estimates of the three pollutants, their age and sex specific estimates were also not significant and are presented in the Appendix A.

Season specific results presented in Figure 3 reveal that associations of CVD mortality with PM_10_, NO_2_, SO_2_, and O_3_ are present in the cold and wet period only. This seasonal interaction reached statistical significance (*p*-value of 0.04) in case of SO_2_. In contrast, results for RD remained rather similar across seasons with no material differences.

Figure 4 illustrates the effects of a 10 µg/m^3^ increase in the respective pollutants on CVD mortality across lags 0 to 21. These models reveal rather similar lag patterns for PM_10_, NO_2_ and SO_2_, with positive estimates for the very first days but negative associations from approximately day three to day 15, indicating the presence of harvesting. In case of PM_10_, estimates are significantly negative across several lags. A similar observation is noted for RD mortality and PM_10_ (see Figure 5), with negative and significant estimates between lag 5–10. Here, lag 0–1 estimates for NO_2_ reached statistical significance and slightly negative estimates were observed only for lags 10 to 14. However, the graph for O_3_ is difficult to interpret given the troughs, whereas SO_2_ is somewhat similar showing a non-significant immediate effect.

## 4. Discussion

### 4.1. Brief Overview

In this study, we estimated short-term associations between ambient air pollutants (PM_10_, NO_2_, SO_2_, and O_3_) and deaths due to cardiovascular and respiratory diseases through separate single- and multiple-pollutant models, overall and by age group, sex and season, in Cape Town, South Africa. It is noteworthy that the air pollution level in the present study are similar to those reported in North America and Europe but stronger associations were detected. We found that PM_10_, NO_2_, SO_2_, and O_3_ were all associated with CVD death risk associations and were statistically significant during the wet and cold season but absent during the warm and dry season. Associations with PM_10_ remained at least marginally significant across all multipollutant models, whereas those for NO_2_ and O_3_ were more sensitive to the adjustment for other pollutants. Results of SO_2_ collapsed in all models involving more than two pollutants. Conversely, only NO_2_ showed prominent and robust associations with RD mortality. For PM_10_, NO_2_ and SO_2_, associations for age ≥ 65 years were slightly stronger than those for age 15 to 65 years. The role of gender in modifying estimates was conclusive only for O_3_ on RD deaths, with significantly higher risk among men.

### 4.2. Cardiovascular Disease and Air Pollutants

Cardiovascular causes of death are usually due to a group of cardiovascular disorders affecting the heart and blood vessels, which may be the underlying cause of acute cardiovascular events leading to hospitalization and death. These acute events include cerebrovascular or pulmonary embolism or myocardial infarctions due to coronary heart diseases, to name a few [23]. There are many risk factors triggering acute cardiovascular events, including exposure to air pollution. Nawrot et al. calculated population attributable fractions for air pollution and reported that a 10 µg/m^3^ change in PM_10_ was associated with a 1.6% (0.9–2.4%) change in incidence of myocardial infarction (MI) [24]. Additionally, interaction between physical exertion and ambient air pollution or temperature further increases the risk of MI [24]. In the present study, a 10 µg/m^3^ rise in daily mean PM_10_ concentrations was associated with a 1.2% to 1.8% increase in daily mortality. These estimates are at least two times higher than what has been reported for single-pollutant associations in systematic reviews and large multi-city studies [5,13,25,26,27]. Although the estimate for PM_10_ decreased from 1.5% to 1.1% after adjustment for NO_2_, it remains larger than expected from the literature. Similarly, NO_2_ was also positively associated with CVD mortality. In comparison to multi-city studies, our estimates are at least five times higher than what the same authors reported for a 10 g/m^3^ increase in NO_2_ [28,29], and two times higher than in another study [25]. However, estimates produced from a systematic review restricted to results from low- and middle-income countries (LMIC) reported an increase of 1.7% per 10 µg/m^3^ NO_2_ [31] which is somewhat comparable to our findings. In addition, an Italian multi-city study reported an increased risk of 2.6% for cardiac mortality. This is compatible with our single pollutant estimate (2.1%), although their analysis used longer lags (0–5) and was restricted to ages over 35 [32]. Interestingly, the significant positive association observed in our study is smaller as compared to earlier findings from the same study area which used data from 2001 to 2006. The authors reported a 3.4% increased risk per IQR rise of 12 µg/m^3^ NO_2_ compared to our 2.2% increased risk per 11 µg/m^3^ [7]. We observed a comparable association of 3.2% per 11 µg/m^3^ only in models additionally containing O_3_ and SO_2_. Therefore, our results show evidence of a persisting increased CVD mortality risk due to NO_2_ in the City of Cape Town over a period spanning 15 years.

Similarly to other studies, Cape Town results for PM_10_ and NO_2_ appear not to be fully independent, but partly capture overlapping characteristics of the two pollutants. In all CVD models which included both PM_10_ and NO_2_, the respective estimates were attenuated—more strongly in case of NO_2_—where confidence intervals included values consistent with the null hypotheses. As discussed in cases of similar previous findings, this attenuation might reflect the correlation of the two markers of ambient air pollution sharing similar sources and perhaps health relevant seasonal patterns [13].

For SO_2_, the significant positive association (risk increase = 2.3%, 95% CI: 0.1–4.7%) per 10 µg/m^3^ was limited to the single pollutant model, with no evidence of independent associations, particularly after co-pollutant adjustments with PM_10_ and O_3_. Our result is similar to the one of an earlier study [7] but is at least three times higher than pooled estimates from LMIC studies [31], two times higher than single and multiple-pollutant meta-analysis estimates for North America and Europe [33], and five times higher than a respective estimate for all-cause mortality [6].

We found both positive and independent associations between O_3_ and CVD mortality with a risk increase of 1.6% for a 10 µg/m^3^ increment in the two-day mean of O_3_. The magnitude of this association is at least four times larger than what has been demonstrated in LMIC, North America, European and global systematic reviews [5,30,31,34]. Furthermore, when compared to a multi-city study including a South African province, our estimate is six times higher than what was reported per 10 µg/m^3^ increase in O_3_ [14]. On the other hand, consistent with our results, an Italian multi-city study found an increased risk of 2.3% (95% CI: 1.1–3.5%) which remained unchanged when adjusted for PM_10_ [35]. In contrast to an Asian multi-city study where the average concentrations of all four pollutants were at least twice the levels we observed in Cape Town, our CVD mortality estimates are at least twice what they reported per 10 µg/m^3^ increase in each pollutant [36].

It is an interesting finding that all of our single-pollutant associations between the four pollutants and CVD mortality are consistently higher than what has been reported in the literature. A few things might be responsible for this unexpected observation. Firstly, South Africa is one of the most industrialized countries in the Southern hemisphere, with significant mining and metallurgical activities. It generates around 91% of its energy from coal burning [37]. In addition, as an arid country, it has high natural dust levels, which adds to the particles from industrial and vehicular emissions. This, compounded with smoke from residential coal combustion in lower-income urban communities, was reported to contribute as much as 30% of the particulate pollution in the country [38]. The pollutants’ profiles and sources could modify the toxic properties—including the oxidative potential—of the air pollution mixture in Cape Town, which may not be fully captured by a purely mass-based characterisation of pollution. For example, the coarse fraction of PM_10_ was much more strongly associated with daily mortality during Sahara dust episodes as compared to other days in Europe. It was reported that a daily increase of 10 µg/m^3^ in PM_10_ after adjusting for PM_2.5_ on Saharan dust days increased mortality by 8.4% (95% CI 1.5–15.8%) compared with 1.3% (95% CI −0.8–3.4%) during non-Saharan dust days (*p*-value for interaction = 0.05) [39]. In South Africa, 71% of dust plumes originate from the Free State province when agricultural areas are exposed to wind during drought from June to January, which overlaps with Cape Town’s wet and dry seasons [40].

Given the lack of PM_2.5_ data in our study, we cannot further evaluate whether our larger estimates are driven by the coarse fraction of PM_10_ or whether the fine PM_2.5_ fraction is more toxic in Cape Town then elsewhere. Source apportionment of daily levels of PM_2.5_ could further elucidate the reasons behind the stronger associations observed in our data. PM_2.5_ samples were collected for 121 days in Cape Town, and when analysed Na, Cl^−^,Mg, Ca, Zn, Al, Fe, SO_4_^2−,^ and NO_3_^−^ were detected in the samples. This indicates that the components found in the PM_2.5_ samples are from multiple sources such as traffic, biomass fuel burning, soot, local road dust and natural sources such as sea sand, soil and mineral dust. Some of these components have been linked to adverse health effects such as SO_4_^2−,^ and NO_3_^−^ which were reported to be positively associated with all-cause mortality with an increased risk of 0.15% and 0.17% per 1 µg/m^3^, respectively [41]. However, those PM_2.5_ speciation data are not sufficient to compare or qualify the toxicity of PM_2.5_ in Cape Town as compared to elsewhere [16]. PM_2_._5_ times-series analyses would be needed to better understand the role of the fine and course fractions of PM in Cape Town.

### 4.3. Respiratory Disease

Deaths from acute respiratory diseases (RD) are typically related to various acute exacerbations of lung diseases due to e.g., respiratory infections, including pneumonia, or asthma attacks. Acute deterioration of lung diseases can be caused by various ambient air pollutants. We found a statistically significant positive association with RD mortality only for NO_2_ and a non-significant positive association with SO_2_. This is in contrast with PM_10_ and O_3_, where the associations were virtually absent. Positive associations of RD-mortality with NO_2_ have been reported in other studies [5,31,42], although estimates were not quite as large as ours.

The null findings for PM_10_ and O_3_ for RD mortality in our study are inconsistent with the existing literature. For instance, Orellano et al. found a 0.91% (95% CI: 0.6–1.2%) increased risk for a 10 µg/m^3^ rise in PM_10_, and Newell et al. reported a corresponding estimate of 0.38% (95% CI: 0·33–0·43). Our results are also consistent with other findings from Cape Town Wichmann et al. [7] and Thabethe et al. [15] reported non-significant positive associations of PM_10_ with daily RD mortality rates as well. The same studies also reported results similar to ours for SO_2_. Large multicity studies and systematic reviews demonstrated significant positive associations between SO_2_ and RD-mortality [6,31]. Our null finding for O_3_ and RD-mortality is consistent with the result from a systematic review involving studies from low- and middle income countries having found statistically non-significant positive associations between RD-mortality and O_3_, with a risk increase by 0.26% (95% CI: −0.09–0.61) for a 10 µg/m^3^ increase in Ozone.

It might be that, in Cape Town, the exposure to features of air pollution relevant to respiratory diseases is better reflected by NO_2_ than by the other pollutants due to specific sources and different population behavioural factors [43]. The strong and independent association with NO_2_ could indicate that it acts as a marker for specific mixtures of pollutants not well captured by PM_10_, SO_2_ or O_3_, e.g., those particularly generated by vehicle exhausts including fine or ultrafine particles. In addition, the association between NO_2_ and RD mortality may reflect, to a larger extent, the effects from sources other than traffic such as power plants [29].

### 4.4. Effect Modification by Sex

When we stratified the analysis by sex we found a statistically significant difference (*p*-value = 0.01) in the association between O_3_ and RD mortality only, with a higher risk in men. Shin et al. found a higher risk in females than males for O_3_ using data covering 52% of the Canadian population [44]. They also reported females to be at higher risk due to NO_2_ exposure, which is consistent with our findings. Bell et al. conducted a meta-analysis using nine ozone-RD mortality studies and concluded that the evidence of a higher association in females than males is limited or suggestive [45]. Other studies have also reported inconsistencies for sex-specific health risk estimates associated with air pollutants [35]. The minor sex-differences in our observed relative risks of CV-mortality with stronger associations among females for PM_10_ and NO_2_ but stronger associations among males in the case of SO_2_ and O_3_ might be chance findings.

### 4.5. Effect Modification by Age

We did not find statistically significant effect modification by age. However, the association between all four pollutants and CVD mortality was consistently higher in age ≥65 years and the positive associations for age 15–64 years did not reach statistical significance. In contrast, RD mortality risk estimates were higher among the elderly only for NO_2_ and SO_2_. Older people showing stronger sensitivity to ambient pollution is well demonstrated in the literature [33,35,36,46,47]. This stronger association among the elderly is plausible given that, according to the 2019 Western Cape provincial burden of disease report, the most common causes of death among the elderly were CVD and other non-communicable disease (NCD) while people younger than 65 years die more from HIV/TB and intentional injuries [48]. This is evident in our study as the number of deaths due to CVD for the elderly being1.7 times higher than in the 15–64 age group. In addition, the elderly are likely to have co-morbidities with reduced immune system and physiological changes, which may lead to decreased cardiac output, increased blood pressure, and the development of arteriosclerosis [46]. Zeka et al. demonstrated that the risk of mortality for PM_10_ doubled by the presence of a secondary diagnosis such as pneumonia, stroke, diabetes, and heart failure [47]. There is evidence of high polymorbidity in South Africa, as 47% of the deaths reported between 1997 and 2017 had more than one cause [49]. Therefore, exposure to increased levels of air pollution may have advanced death events among frail individuals. This is suggested by the lag structure of PM_10_-effects on CVD and RD mortality (Figure 4 and Appendix A) showing harvesting with significant negative effects from lag 5–10. The pattern is even more pronounced among the elderly than when combining all age groups, suggesting that harvesting mostly occurred in the elderly.

### 4.6. Effect Modification by Season

The health effect estimates during the cold/wet season were higher than in the warm/dry season, but this difference was only statistically significant for SO_2_. Our findings are not in agreement with evidence shown in multiple studies [45].

Results from other South African studies corroborate our findings of elevated pollution levels in the cold/wet season [50,51,52], as shown in Table 1. This is likely due to season-specific differences in anthropogenic activities and environmental and meteorological factors being associated with changes in sources, composition and toxicity of the pollutants

### 4.7. Harvesting of Frailty by Air Pollution

In contrast to many studies of the acute effects of ambient air pollution, our results for CVD and RD provide rather strong evidence for harvesting [53]. The concept of harvesting assumes that air pollution may advance death by just a few days among particularly frail people who are at a higher risk of dying at any given time. Under such a model, days with increased death rates due to air pollution would be followed by death rates falling below the expected averages as the pool of frail people had been reduced or “harvested” by the acute effects of air pollution. Several studies have investigated harvesting by air pollutants for different health outcomes [53,54,55,56,57,58,59,60]. Of these, only one has reported significant results, Costa et al. found evidence of mortality displacement for non-accidental mortality associated with PM_10_ among the elderly across lags 0–30, while others have used lags up to 45 days with no evidence for acute effects being fully compensated by harvesting [53]. Schwartz [59] and Zanobetti [56] described that subsequent to the harvesting period where negative effects are observed, there should be a rebound of risk estimates as a consequence of the replenishment of the risk pool. In our case, the period of three weeks may not have been long enough for this. Nonetheless, for CVD mortality we see prevailing short-term effects and considerable amount of mortality displacement in the curves for PM_10_, NO_2_ and SO_2_ particularly with significant negative effects in PM_10_. Additionally, there is indication of delayed acute effects with the rebound to positive estimates from lag 15 onward, for all the pollutants except O_3_. This is more visible for PM_10_ when the lags are extended to 45 days (Appendix A).

On the other hand, for RD mortality, there is more evidence of delayed associations with NO_2_ exposure for all ages (as shown in Figure 5) up until lag 8, while SO_2_ and O_3_ curves remain positive over the entire lag period (RR >1). Instead, PM_10_ associations for the first few days are statistically non-significant, while lags 5–10 indicate some harvesting with no signs of rebounds later on. Furthermore, it is important to note that the very short term exposures to all four pollutants at lags 0 and 1 had the strongest association with CVD-mortality.

## 5. Strengths and Limitations

This study has several strengths. First, we examined the association of four important air pollutants with cardiovascular and respiratory mortality using a multipollutant approach. Secondly, this study benefitted from a relatively low correlation among the pollutants which enabled us to obtain relatively stable effect estimates in multipollutant models. Thirdly, the mortality data was collected from the national mortality database for the City of Cape Town, which allows for the generalization of the findings to the study area. Finally, our results contribute to the increasing body of evidence that support the independent health effects of ozone on cardiovascular mortality and of NO_2_ on respiratory mortality.

We also acknowledge some limitations of this study. Exposure misclassification cannot be ruled out, as average daily concentration levels of the pollutants obtained from measurements at fixed monitoring sites may not be fully representative of the average exposure across the whole population of the city. While data on PM_10_ and NO_2_ were quite complete, measurements of SO_2_ were missing on 24% of the days, and O_3_ measurements were missing for 29% of the study period. Therefore, multi-pollutant models including O_3_ used fewer observations, for instance 62% of the data for four-pollutant models. However, when we restricted the single pollutant modelling of PM_10_, NO_2_ and SO_2_ to days where ozone data were available, this shrunk the sample size by at least 1000 days. We observed that the effect estimates for PM_10_ (RR = 2.6%, 95% CI 0.7–4.4%) and NO_2_ (RR = 2.7%, 95% CI 0.5–5.0%) increased slightly, while those for SO_2_ were reduced (RR = 0.6%, 95% CI −1.3–2.5%), resulting in a wider confidence interval due to the smaller sample size. This suggests that environmental conditions on days with missing O_3_-data were slightly different from those on days with O_3_-data. In addition, there was a lack of information on individual characteristics that would help us understand the stronger associations between air pollution and mortality when compared to other settings.

## 6. Conclusions

This study advances our understanding of the mortality risk associated with short-term exposure of multiple air pollutants (PM_10_, NO_2_, SO_2_, and O_3_) in Cape Town, South Africa. We found evidence of a positive association between the four pollutants and CVD-mortality, but only NO_2_ was significantly associated with RD mortality. In addition, CVD-mortality among the elderly was advanced by a few days due to short-term PM_10_ exposure. Additional research is needed to confirm our findings, particularly in regard to better quality data and PM_2.5_

## Figures and Tables

**Figure 1 ijerph-19-08078-f001:**
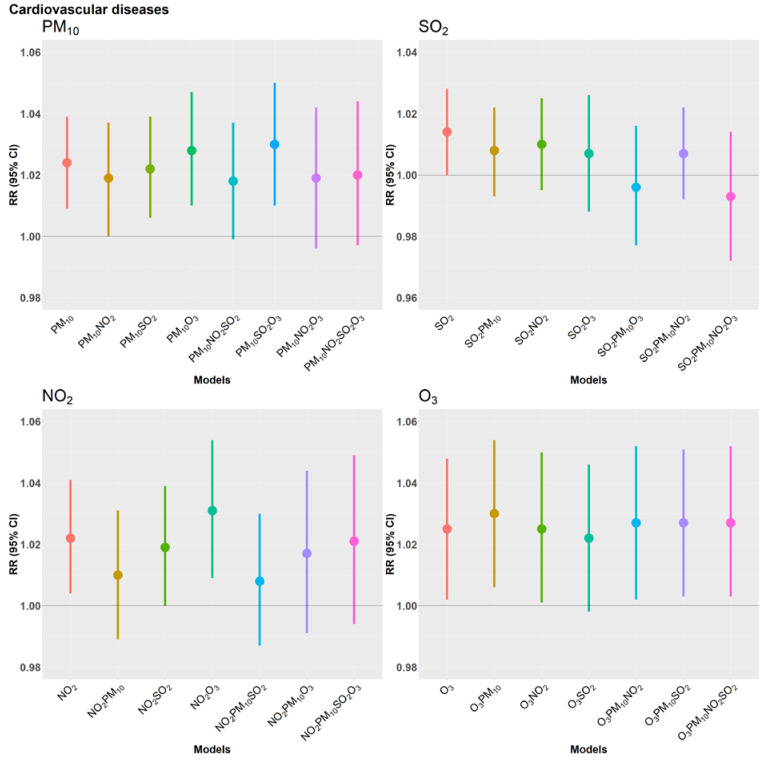
Overall relative risks estimated from Quasi-Poisson regression models of daily counts of cardiovascular disease deaths, adjusting for time trends and seasonal variation, day of the week, public holiday, temperature and relative humidity. Estimates are presented per IQR µg/m^3^ increase in the two-day moving average (lag 0–1) of PM_10_, NO_2_, SO_2_ and O_3_ for the respective two-, three- and four-pollutant models.

**Figure 2 ijerph-19-08078-f002:**
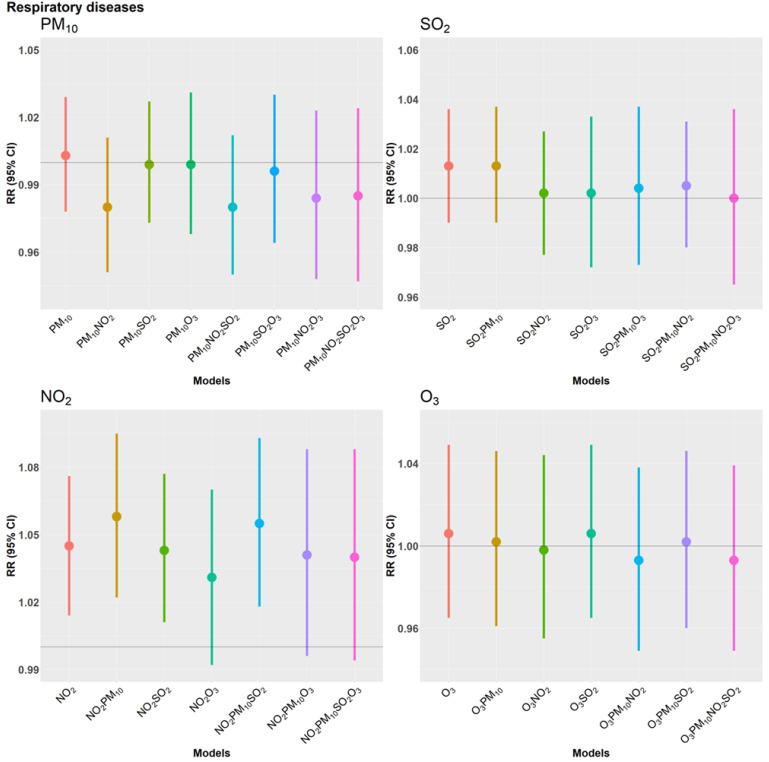
Overall relative risk estimated from Quasi-Poisson regression models of daily counts of respiratory disease deaths, adjusting for time trends and seasonal variation, day of the week, public holiday, temperature and relative humidity. Estimates are presented per IQR µg/m^3^ increase in the 2-day moving average (lag 0–1) of PM_10_, NO_2_, SO_2_ and O_3_ for the respective two-, three- and four-pollutant models.

**Figure 3 ijerph-19-08078-f003:**
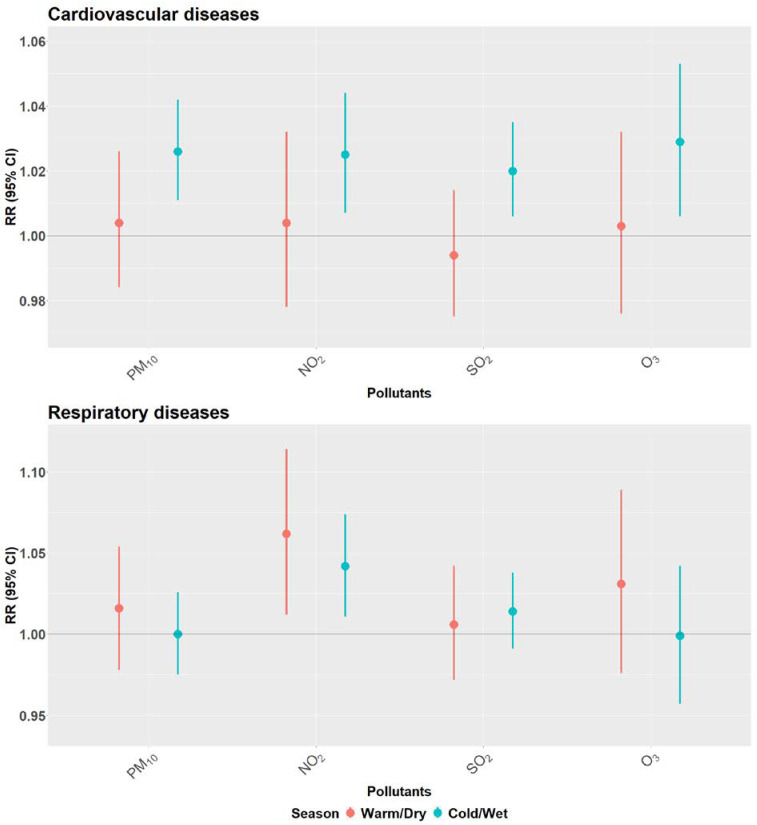
Overall season-specific estimates for daily counts of cardiovascular and respiratory deaths from Quasi-Poisson regression models adjusting for time trends and seasonal variation, day of the week, public holiday, temperature and relative humidity. Estimates are presented per IQR µg/m^3^ increase in the two-day moving average (lag 0–1) of PM_10_, NO_2_, SO_2_ and O_3_.

**Figure 4 ijerph-19-08078-f004:**
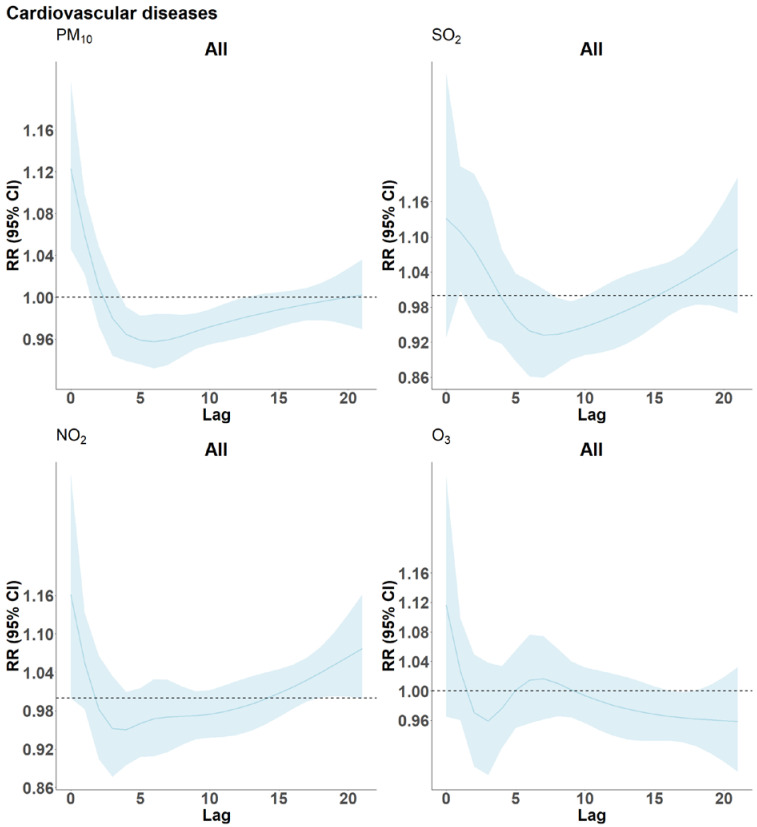
Lag structure (0–21) of the estimated effects of a 10 µg/m^3^ increase in PM_10_, NO_2_, SO_2_ and O_3_ concentrations on cardiovascular disease mortality in Cape Town, South Africa, 2006–2015. The blue curve gives the RR-estimates and the light blue band their 95%-confidence intervals.

**Figure 5 ijerph-19-08078-f005:**
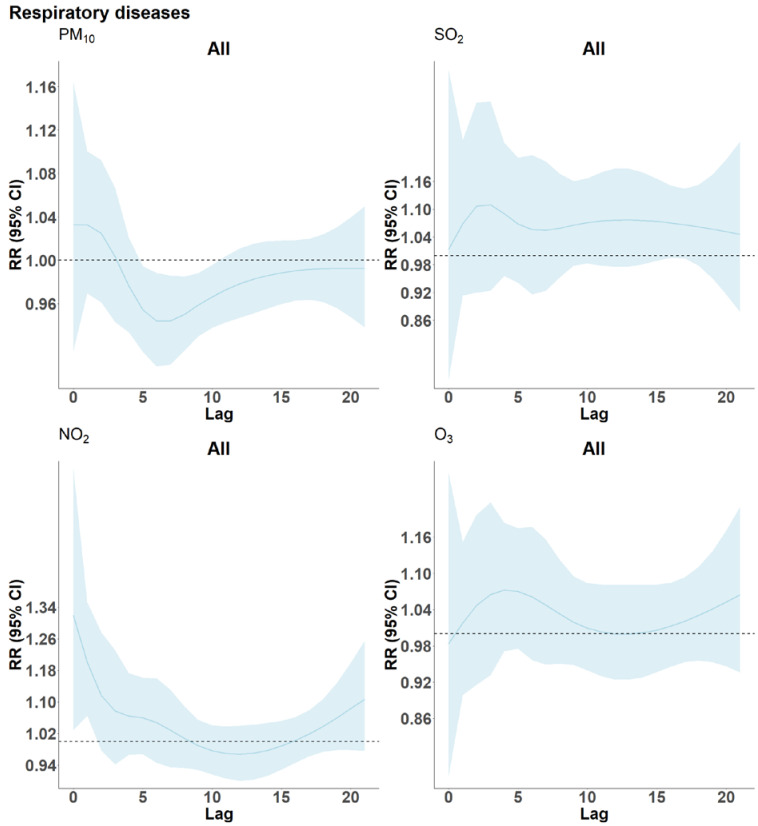
Lag structure (0–21) of the estimated effects of a 10 µg/m^3^ increase in PM_10_, NO_2_, SO_2_ and O_3_ concentrations on respiratory disease mortality in Cape Town, South Africa, 2006–2015. The blue curve gives the RR-estimates and the light blue band their 95%-confidence intervals.

**Table 1 ijerph-19-08078-t001:** Descriptive statistics for daily counts of cardiovascular and respiratory mortality, daily concentrations of PM_10_, NO_2_, SO_2_ and O_3_ and daily meteorological variables in Cape Town, South Africa during 1 January 2006 to 31 December 2015.

						Percentiles			By Season-Mean (SD)	
Variable	Mean	SD	Min	Max	IQR	25th	50th	75th	Warm/Dry	Cold/Wet
Cardiovascular disease									n = 33,427	n = 20,929
All ages and sex	14.88	4.49	2	33	6	12	15	18	13.80 (4.01)	17.01 (4.61)
n = 54,356										
15–64 years	5.52	2.50	0	18	3	4	5	7	5.25 (2.37)	6.05 (2.67)
n = 20,145										
>65 years	9.35	3.49	0	26	5	7	9	12	8.54 (3.12)	11 (3.61)
n = 34,164										
Female	7.70	3.01	0	20	4	6	7	10	7.14 (2.78)	8.81 (3.13)
n = 28,133										
Male	7.17	2.90	0	23	4	5	7	9	6.65 (2.68)	8.19 (3.04)
n = 26,167										
Respiratory disease									n = 11,625	n = 8751
All ages and sex	5.58	2.85	0	75	3	4	5	7	4.80 (2.36)	7.11 (3.09)
n = 20,376										
15–64 years	2.67	1.78	0	32	3	1	2	4	2.31 (1.55)	3.36 (1.98)
n = 9735										
>65 years	2.90	1.93	0	17	3	1	3	4	2.47 (1.67)	3.74 (2.14)
n = 10,588										
Female	2.48	1.74	0	44	2	1	2	3	2.12 (1.53)	3.19 (1.91)
n = 11,270										
Male	3.09	1.94	0	40	2	2	3	4	2.67 (1.68)	3.90 (2.18)
n = 9063										
Air pollutants										
PM_10_ (µg/m^3^)	30.35	13.6	6.56	98.63	16.35	20.44	27.32	36.8	29.5 (12.33)	31.53 (15.73)
n = 3647										
NO_2_ (µg/m^3^)	16.63	8.81	2.63	59.24	10.66	10.47	14.77	21.13	14.57 (7.61)	20.33 (9.58)
n = 3603										
SO_2_ (µg/m^3^)	10.54	5.46	2.31	49	6	6.88	9.20	12.87	9.81 (4.78)	12 (6.35)
n = 2769										
O_3_ (µg/m^3^)	33.06	12.28	2.38	89.08	15.60	24.87	33.06	40.46	32.35 (12)	34.55 (12.7)
n = 2672										
Meteorological data										
Temperature (°C)	17.39	11.02	30.67	100	15.67	62	70.3	77	19.3 (3.49)	13.6 (2.33)
Relative humidity (%)	69.8	4.2	7.5	30	6.8	14	17.3	20.7	67 (10)	76 (10.8)

Abbreviations: SD—standard deviation; Min—minimum; Max—maximum; IQR—interquartile range. Warm/dry period: November–March; Cold/wet period: April–October.

**Table 2 ijerph-19-08078-t002:** Spearman’s rank correlation between city-level daily mean PM_10_, NO_2_, SO_2_, O_3_ and meteorological parameters during the period of 1 January 2006 to 31 December 2015 in Cape Town, South Africa.

	PM_10_	NO_2_	SO_2_	O_3_	Temperature	Humidity
PM_10_	1	0.38	0.27	0.12	0.17	−0.25
NO_2_		1	0.40	0.06	−0.35	0.07
SO_2_			1	−0.11	−0.12	−0.01
O_3_				1	−0.13	0.01
Temperature					1	−0.41
Humidity						1

**Table 3 ijerph-19-08078-t003:** Relative risk estimated from Quasi-Poisson regression models of cardiovascular and respiratory disease deaths, adjusting for time trends and seasonal variation, day of the week, public holiday, and meteorological factors including temperature and relative humidity. Estimates are presented per IQR increase in the two-day moving average (lag 0–1) of PM_10_, NO_2_, SO_2_ and O_3_ for all age and sex groups.

Cardiovascular Disease Deaths by Pollutants												
Per IQR µg/m^3^	Per 16 µg/m^3^ PM_10_			Per 11 µg/m^3^ NO_2_			Per 6 µg/m^3^ SO_2_			Per 16 µg/m^3^ O_3_		
Groups	RR	95% Confidence Interval		RR	95% Confidence Interval		RR	95% Confidence Interval		RR	95% Confidence Interval	
		Lower	Upper		Lower	Upper		Lower	Upper		Lower	Upper
All	1.024	1.009	1.039	1.022	1.004	1.041	1.014	1	1.028	1.025	1.002	1.048
Age 15–64	1.011	0.986	1.036	1.01	0.981	1.041	1.01	0.988	1.032	1.019	0.982	1.057
Age ≥65	1.033	1.014	1.052	1.031	1.008	1.054	1.017	0.999	1.034	1.029	1	1.059
Female	1.032	1.012	1.053	1.026	1.001	1.051	1.007	0.989	1.026	1.018	0.987	1.05
Male	1.015	0.994	1.037	1.019	0.993	1.045	1.022	1.003	1.041	1.035	1.002	1.069
**Respiratory disease deaths by Pollutants**												
**Per IQR µg/m^3^**	**Per 16 µg/m^3^ PM_10_**			**Per 11 µg/m^3^ NO_2_**			**Per 6 µg/m^3^ SO_2_**			**Per 16 µg/m^3^ O_3_**		
**Groups**	**RR**	**95% Confidence Interval**		**RR**	**95% Confidence Interval**		**RR**	**95% Confidence Interval**		**RR**	**95% Confidence Interval**	
		**Lower**	**Upper**		**Lower**	**Upper**		**Lower**	**Upper**		**Lower**	**Upper**
All	1.003	0.978	1.029	1.045	1.014	1.076	1.013	0.99	1.036	1.006	0.965	1.049
Age 15–64	1.003	0.969	1.039	1.041	0.997	1.085	1.01	0.979	1.042	1.007	0.951	1.067
Age ≥65	1.003	0.969	1.038	1.049	1.007	1.093	1.016	0.985	1.047	1.004	0.948	1.063
Female	1.009	0.973	1.047	1.049	1.004	1.096	1.021	0.988	1.055	0.949	0.894	1.008
Male	1	0.967	1.033	1.042	1.002	1.083	1.005	0.976	1.035	1.054	0.998	1.113

## Data Availability

Exposure data are available for download on the South African Air Quality Information System (SAAQIS) website; however, restrictions apply to the health outcome data.

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
