# Peer review of "Short-Term Effects of PM10, NO2, SO2 and O3 on Cardio-Respiratory Mortality in Cape Town, South Africa, 2006–2015"

_ijerph, 2022, doi:10.3390/ijerph19138078_

Round 1

Reviewer 1 Report

In this paper, a comprehensive of the DLNM model was used in Cape Town, South Africa, the Short-Term impact of PM10, NO2, SO2 and O3 on Cardio-Respiratory Mortality are assessed. These studies aim to bridge this gap by estimating the associations of air pollution with daily mortality due to respiratory and cardiovascular diseases combine with single and multi-pollutant models,which is helpful to comprehend the associations between air pollution and mortality in this region.

Some moments should be clarified in the article in more detail.

Introduction

For instance, PM10 is a marker of traffic emissions in addition to other combustion and non-combustion sources. In many cities, NO2 is a marker of traffic pollution, while SO2 may point to power plant emissions and other fossil fuel combustion sources. Modify the subscript.

Discussion

Suggest to combine the sex, age and season paragraphs to avoid logical confusion. Please read the full revision carefully.

Also, the specific significance of the research should be added to the manuscript.

Author Response

We will like to thank the reviewer for taking the time and effort to review our manuscript, we greatly appreciate the comments provided.

Point 1: Introduction

For instance, PM10 is a marker of traffic emissions in addition to other combustion and non-combustion sources. In many cities, NO2 is a marker of traffic pollution, while SO2 may point to power plant emissions and other fossil fuel combustion sources. Modify the subscript.

Response: The subscripts in this text and other have been modified

Point 2: Discussion

Suggest to combine the sex, age and season paragraphs to avoid logical confusion. Please read the full revision carefully.

Also, the specific significance of the research should be added to the manuscript.

Response: We believe the current layout of the paragraphs avoids logical confusion as there are sub-headings which reflects the texts. This also helps the reader navigate easily to the section that interests them the most. 

The significance/importance of the research was well addressed, particularly discussing the rarity of such research in Africa in comparison to Europe and North America. Perhaps there is something in particular that the reviewer will like us to address and we will be happy to if this can be provided in a little more detail.

Reviewer 2 Report

The authors should be congratulated for their study on the short-term effects of air pollution on cardiac and respiratory mortality in Cape Town. The authors analyzed the influence of many pollutants (also in multi-pollutant models which is rare) over a long period of time 2006-2015. The other strengths of the study include: 1) further confirmation of the role of ozone on cardiovascular mortality and N02 on respiratory mortality on the large group and 2) identification of the harvesting for PM10 (deaths in the frail population in the initial days of increased pollution). The study is overall well written and informative. However, I have some comments, which may improve the manuscript:

1. Did you considered including atmospheric pressure or large events (such as sport events - World Cup in football or rugby) as potential confounders? Do you think those could influence the results (increase CV mortality as demonstrated by some studies), did those events coincide with periods of increased pollution?

2. In the methods section please add information on the total size of Cape Town population - I did not find this information. 

3. Page 11 - first sentence under the figure - there is an "error!..." instead of a reference to a figure 

4. I would change the first sentence of the abstract and parts of the introduction to sound that health effects of air pollution are rarely studied in Africa, not sub-Sahara Africa as most of the studies come from Cape Town. In the current version it sound contradictory. 

5. Please explain IQR and harvesting (or use other term) in the abstract and on the first mention in the text. A reader unfamiliar with the terminology may not know that harvesting is the advancement of death in the frail population in the first days of air pollution. 

6. Please explain why you have decided to exclude children <15 years of age from the analysis. 

Author Response

We will like to thank the reviewer for taking the time and effort to review our manuscript, we greatly appreciate the comments provided.

Point 1. Did you considered including atmospheric pressure or large events (such as sport events - World Cup in football or rugby) as potential confounders? Do you think those could influence the results (increase CV mortality as demonstrated by some studies), did those events coincide with periods of increased pollution?

Response 1: we are aware that research has shown watching sports either on television or at the stadium can increase viewers' risk of CVD related illness particularly if they have existing conditions. However, during the planning of this study we did not consider collecting sports related data such as games and their viewing dates. In hindsight, it could have been informative to control for this in the models given how South Africa is passionate about soccer and rugby, thus, it will be considered for future studies.

Point 2: In the methods section please add information on the total size of Cape Town population - I did not find this information. 

Response 2: Thanks for highlighting this, a paragraph has been added from line 115-120 addressing it.

Point 3: Page 11 - first sentence under the figure - there is an "error!..." instead of a reference to a figure 

Response 3: The manuscript downloaded did not show an error, the figure and reference works fine therefore there was nothing to fix.

4. I would change the first sentence of the abstract and parts of the introduction to sound that health effects of air pollution are rarely studied in Africa, not sub-Sahara Africa as most of the studies come from Cape Town. In the current version it sound contradictory. 

Response 4: this comment was well-received and sub-Sahara Africa has been replaced with Africa throughout the manuscript.

5. Please explain IQR and harvesting (or use other term) in the abstract and on the first mention in the text. A reader unfamiliar with the terminology may not know that harvesting is the advancement of death in the frail population in the first days of air pollution. 

Response 5: Given the limited word count for the abstract, we wrote out "Interquartile range" and changed "harvesting" to "mortality displacement" in the abstract. 

Point 6: Please explain why you have decided to exclude children <15 years of age from the analysis.

Response 6: We briefly addressed this in line 127-128. The statistics office did not give us death for under 15 and there was no explanation provided for this.  We think it might be due to general data protection regulation (GDPR). In addition, it took more than a year to obtain the data, so we used what we were given